# Nationwide seroprevalence of hepatitis A in South Korea from 2009 to 2019

**Deog-Yong Lee**[1], **Su-Jin Chae**[1], **Seung-Rye Cho**[1], **Wooyoung Choi**[1], **Chang-Ki Kim**[2], **Myung-Guk Han**[1]*

**1** Division of Viral Diseases, Center for Laboratory Control of Infectious Diseases, Korea Centers for Disease Control and Prevention, Osong, Republic of Korea, **2** Seoul Clinical Laboratories, Yongin, Republic of Korea

* mghan@korea.kr

## Abstract

Hepatitis A, an acute type of hepatitis caused by the hepatitis A virus, occurs worldwide. Following the 2009 hepatitis A epidemic in South Korea, patient outbreak reports were collectively converted to an "all-patient report" in 2011, and national immunization programs were introduced for children in 2015. In this study, we aimed to analyze the changes and characteristics of hepatitis A antibody titers in South Korea following the epidemic. The results of hepatitis A antibody tests performed at clinical laboratories from 2009 to 2019 were analyzed based on year, age, region, sex, and medical institution. The average 2009–2018 positive anti-hepatitis A virus immunoglobulin G rate was 51.8%, but it increased (56.06%) in 2019. Significantly different antibody-positive rates were observed based on age: <10 years, 54.5%; 20–29 years, 19.5%; ≥50 years, almost 100%. The positive rate of individuals in their teens and 20s gradually increased, whereas that of those in their 30s and 40s gradually decreased. Males had higher antibody-positive rates than females, and samples from higher-level general hospitals exhibited higher antibody rates. The positive anti-hepatitis A virus immunoglobulin M rates gradually decreased after 2009 and were <1% after 2012. However, a high positive rate of 3.69% was observed in 2019 when there was an epidemic. Anti-hepatitis A virus immunoglobulin G-positive rates were similar throughout the year, but the anti-hepatitis A virus immunoglobulin M-positive rates increased from January, peaked in April, and decreased from July, exhibiting distinct seasonality. This is considered to be related to groundwater pollution during the spring drought season. The introduction of the "all-patient report" and national vaccination program for children has had an effective influence on hepatitis A management. However, for hepatitis A prevention, policy considerations for high-risk age groups with low antibody-positive rates will be necessary.

## Introduction

Hepatitis A is an acute form of hepatitis caused by the hepatitis A virus (HAV) and occurs worldwide. The World Health Organization estimates that approximately 1.4 million patients are diagnosed with this disease every year [1]. The HAV has been classified under the *Hepatovirus* genus and the Picornaviridae family and is a 27-nm non-enveloped ribose nucleic acid (RNA) virus. The viral gene consists of 7,500 base pairs and is largely divided into three (P1,

**Data Availability Statement:** All relevant data are within the manuscript and Supporting Information files.

**Funding:** This study was conducted under the fund of the hepatitis A surveillance project (Project No.

4837-301 and 4838-303) managed by the Korea Disease Control and Prevention Agency, KDCA (previously, KCDC).

**Competing interests:** The authors have declared that no competing interests exist.

P2, and P3) protein genes. There are seven genotypes (I [A, B], II, III [A, B], IV, V, VI, and VII), and four of them (I, II, III, and VII) cause human infection [2, 3]. The clinical symptoms of hepatitis A are similar to those of acute hepatitis caused by other viruses and include fever, nausea and vomiting, dark brown urine, loss of appetite, abdominal discomfort, and jaundice. Approximately 70% of children younger than 6 years are asymptomatic; however, with increasing age, jaundice is accompanied by severe symptoms, and complications such as Guillain-Barre syndrome, acute renal failure, cholecystitis, pancreatitis, vasculitis, and arthritis may develop. In addition, the risk may be increased if the patient have other liver diseases such as hepatitis B and C or are infected with a different genotype of hepatitis A [1, 4]. Hepatitis A can spread directly through contact with a patient or indirectly by consuming water or food that has been contaminated with the patient's stool. The most effective measure to prevent hepatitis A is to improve environmental and personal hygiene or to inoculate the population with vaccine. Effective and safe vaccines are currently being used [1, 4, 5].

In South Korea, hepatitis A has been observed to mainly develop in individuals from younger age groups, and in the year 2009 in particular, there were several reports regarding hepatitis A patients in their 20s and 30s [4]. Following an epidemic of hepatitis A in 2009, the patient outbreak reports were converted to an "all-patient report" in 2011, and national immunization programs were introduced for children in 2015 [4]. The number of reports on hepatitis A patients has gradually decreased since the initiation of the "all-patient report," decreasing to 867 cases in 2013; however, it increased to 4,000 patients per year in 2016 and 2017. Moreover, due to the epidemic in 2019, 17,635 cases were reported, which is 4.6 times higher than the average number of cases (3,845) in the previous 3 years [4, 6]. There have been several reports on the seroprevalence of hepatitis A in South Korea related to the epidemic in 2009, but no recent trends have been investigated [7–9].

In this study, we aimed to analyze the nationwide hepatitis A antibody test data in order to investigate the characteristics of hepatitis A antibody changes since the epidemic in 2009. We also sought to generate evidence based on policy-related data in order to assess the effectiveness of the current hepatitis A management measures. These aims were achieved.

## Materials and methods

### Study subjects

The data used in this investigation corresponded to the total number of patients who requested tests to detect the hepatitis A antibody from clinical laboratories, and no distinction was made in terms of whether the condition affecting the patient was hepatitis or whether it involved other symptoms. The types of antibodies were investigated separately in patients examined for anti-HAV immunoglobulin G and M (IgG and IgM, respectively). The results of the hepatitis A tests conducted over 10 years between 2009 and 2018 were analyzed first (first phase of analysis), and data associated with the outbreak in 2019, when there was an epidemic, were subsequently analyzed (second phase of analysis). In the first phase of the analysis, data were obtained from the Seoul Clinical Laboratory (SCL), a specialized inspection agency that accepts test samples from hospitals and provides results. This laboratory works on more than 20% of the test volumes commissioned by other hospitals excluding the volumes tested by higher-level hospitals that operate separate laboratories [7]. In the second phase of the analysis, data were obtained from five major domestic laboratories that are in charge of clinical specimen testing (SCL, Eawon, Samkwang, Green Cross, and Seegene) and were analyzed under the same conditions. The proportion of testing performed by these five laboratories accounted for more than 90% of the testing conducted in the country, minimizing the scope for errors in the data based on the area in charge of each laboratory. During the data extraction process,

data were deemed as duplicate, and therefore excluded, if the patient name, medical record number, medical institution name, and results were the same. A total of 870,865 cases from 2009 to 2018 involving anti-HAV IgG data were received from the SCL for the first phase of the analysis, along with 308,650 cases involving anti-IgM data. A total of 596,245 cases from 2019 involving anti-HAV IgG were received from the five laboratories for the secondary analysis, along with 211,629 cases involving anti-IgM (Table 1).

## Anti-HAV test

The HAV antibody titer tested for the levels of anti-HAV IgG and IgM present in patient blood samples. Automated equipment was used for all evaluations; the Seoul Clinical Laboratory, Samkwang, Green Cross, and Seegene used the Architect i2000 system (Abbott Laboratories, Wiesbaden, Germany), and the Eawon Medical Foundation used the Cobas 8000 e602 (F. Hoffmann-La Roche Ltd., Basel, Switzerland) equipment. The first analysis period from 2009 to 2018 involved the data generated by the Architect system used by the SCL, and the second analysis period in 2019 involved data generated by both the aforementioned devices.

## Statistical analysis

The antibody-positive rates of anti-HAV IgG and anti-HAV IgM were classified and summarized according to test status. We analyzed data by year, age, sex, type of medical institution (clinic, hospital, general hospital, advanced general hospital), and region. The change in the antibody-positive rate by age was calculated based on the 10-year positive rate, and in the case of patients who were 10 years or younger, the change in the IgG antibody-positive rate was calculated in 12-month increments. The changes in the positive rate of anti-HAV IgM based on region were compared by year. The positive rate of anti-HAV IgM and patient reporting rate were compared from 2011 onward, when the complete investigation began in South Korea, and the monthly incidence was compared with the positive antibody rate in 2019. The analysis was conducted on the total number of commissioned tests at each laboratory, and no individual demographic information was evaluated.

## Ethics statement

The study collected test results for patient diagnosis and analyzed them retrospectively. The agency that collected the data is a specialized inspection agency that requests a test from the

**Table 1. Number of samples analyzed during the investigation period and the antibody positive rate.**

| Antibody | Samples | Analysis Period | | | | | | | | | | | | Second | Total |
|---|---|---|---|---|---|---|---|---|---|---|---|---|---|---|---|
| | | First | | | | | | | | | | | | | |
| | | 2009 | 2010 | 2011 | 2012 | 2013 | 2014 | 2015 | 2016 | 2017 | 2018 | Sub-total | | 2019 | |
| IgG | Total | 5,274 | 55,551 | 71,367 | 86,977 | 98,548 | 104,553 | 160,917 | 106,868 | 88,298 | 92,512 | 870,865 | | 596,245 | 1,467,110 |
| | Positive | 2,897 | 28,699 | 35,643 | 42,041 | 50,097 | 51,907 | 86,430 | 57,973 | 45,709 | 49,824 | 451,220 | | 334,244 | 785,464 |
| | (%) | (54.93) | (51.66) | (49.94) | (48.34) | (50.84) | (49.65) | (53.71) | (54.25) | (51.77) | (53.86) | (51.81) | | (56.06) | (53.54) |
| | Negative | 2,377 | 26,852 | 35,724 | 44,936 | 48,451 | 52,646 | 74,487 | 48,895 | 42,589 | 42,688 | 419,645 | | 262,001 | 681,646 |
| | (%) | | | | | | | | | | | | | | |
| IgM | Total | 31,617 | 31,197 | 32,637 | 29,101 | 28,885 | 29,659 | 35,806 | 31,540 | 30,447 | 27,761 | 308,650 | | 211,629 | 520,279 |
| | Positive | 3,646 | 1,477 | 786 | 154 | 98 | 137 | 148 | 313 | 241 | 200 | 7,200 | | 7807 | 15,007 |
| | (%) | (11.53) | (4.73) | (2.41) | (0.53) | (0.34) | (0.46) | (0.41) | (0.99) | (0.79) | (0.72) | (2.33) | | (3.69) | (2.88) |
| | Negative | 27,971 | 29,720 | 31,851 | 28,947 | 28,787 | 29,522 | 35,658 | 31,227 | 30,206 | 27,561 | 301,450 | | 203,822 | 505,272 |
| | (%) | | | | | | | | | | | | | | |

hospital. These institutions only received information from patients coded with the sample. These institutions do not have the information to track individual patients. This is because all patient information is provided fully anonymized to the agency. All studies were approved by the formal research ethics committees (Seoul Clinical Laboratories, IRB-19-013; Seegene Medical Foundation, SMF-IRB-2019-009; EONE Laboratories, 128477-202009-BR089; Green Cross Laboratories, 2019–1029; Samkwang Medical Laboratories, S-IRB-2019-009-11-13). The ethics committee did not require an informed consent. Thus, a written informed consent was not provided by the patients. Data that did not fit the research purpose were excluded according to each institution's research ethical standards. Furthermore, no additional information was collected

## Results

### The changing pattern of anti-HAV IgG seroprevalence from 2009 to 2019

In this study, we analyzed 870,865 valid anti-HAV IgG results obtained over the 10-year period from 2009 to 2018, termed the first analysis period. The average positive rate was 51.8%, and no annual increase or decrease was observed (Table 1). There was a significant difference in the antibody-positive rate based on age. The rate was 54.5% in patients younger than 10 years; the lowest positive rate of 19.5% was observed in patients in their 20s, and an almost 100% positive rate was found in those in their 50s and older. The antibody-positive rate was reclassified based on year to analyze the data further. The positive rate in children younger 10 years was 62.1% in 2009, and it increased continuously to 68.1% in 2018. The positive rate in teenagers and individuals in their 20s also gradually increased by year between 2009 and 2018. In teenagers, the rate increased from 22.3% to 53.0%, and in individuals in their 20s, the rate increased from 11.3% to 27.6%. Conversely, we were able to confirm that the positive rate decreased between 2009 and 2018 in patients in their 30s and 40s. In patients in their 40s, the rate decreased from 88.5% to 67.5%, and in patients in their 30s, it decreased from 48.1% to 30.2% (in 2013) and then increased to 32.4% (Fig 1A). The positive IgG rate in patients younger than 10 years was analyzed by further dividing age into 12-month increments. Although the number of samples was small, the positive rate tended to increase from 12 to 24 months of age. In particular, a high positive rate of over 90% was observed in children aged 24–36 months and older from 2015 onward (Table 2). Regarding the changes according to sex, males had higher antibody-positive rates than females (60.6% vs. 47.1%), and regarding changes according to institution type, general hospitals had higher positive rates than clinics or hospitals (77.8%, 50.9%, and 46.0%, respectively). Although there was not much regional variation in the positive rate, it was greater than 60% in Gwangju, South Jeolla, Jeju, Ulsan, South Gyeongsang, and Sejong.

During the second analysis period (2019), 596,245 valid results were analyzed. The total positive rate was 56.06%, which was 4.26% higher than the average positive rate over the previous 10 years (Table 1). The positive rates of children younger than 10 years (88.9%), teenagers (61.7%), and individuals in their 20s (32.8%) were higher than the average and tended to increase continuously. The positive rates of individuals in their 30s (32.4%) and those in their 40s (63.2%) were at the level of the previous year or were slightly decreased. The positive rate was close to 100% in individuals in their 50s and older (Fig 1A). The positive antibody rate was consistently higher in males (60.0%) than in females (53.2%), and among institution type, general hospitals, as in 2009–2018, had the most elevated antibody-positive rates (66.5%). To further analyze this, the positive rates were reclassified by year.

### The changing pattern of anti-HAV IgM seroprevalence from 2009 to 2019

The number of tests analyzed during the first analysis period was 308,650. In the Architect i2000 system (Abbott Laboratories, Wiesbaden, Germany), the defined standard for anti-HAV

**A**

| Year | Age groups | | | | | | | | | Total |
|------|------|-------|-------|-------|-------|-------|-------|-------|------|-------|
|      | 10< | 10-19 | 20-29 | 30-39 | 40-49 | 50-59 | 60-69 | 70-79 | 80≤ |       |
| 2009 | 62.1% | 22.3% | 11.3% | 48.1% | 88.5% | 98.9% | 100.0% | 100.0% | 100.0% | 54.9% |
| 2010 | 59.9% | 17.1% | 10.1% | 39.5% | 88.5% | 99.0% | 99.6% | 98.5% | 91.8% | 51.7% |
| 2011 | 60.6% | 25.1% | 12.4% | 32.7% | 86.7% | 98.8% | 99.5% | 99.2% | 93.7% | 49.9% |
| 2012 | 65.7% | 29.6% | 14.6% | 31.2% | 84.3% | 98.4% | 99.6% | 99.7% | 92.2% | 48.3% |
| 2013 | 61.4% | 33.6% | 16.6% | 30.2% | 81.4% | 98.0% | 99.5% | 99.9% | 96.3% | 50.8% |
| 2014 | 65.6% | 37.1% | 19.1% | 30.3% | 79.6% | 98.1% | 99.6% | 99.8% | 96.4% | 49.6% |
| 2015 | 65.5% | 40.3% | 20.8% | 31.6% | 75.9% | 97.2% | 99.6% | 99.9% | 96.2% | 53.7% |
| 2016 | 70.9% | 44.8% | 21.3% | 30.2% | 74.5% | 96.7% | 99.8% | 99.8% | 98.8% | 54.2% |
| 2017 | 70.9% | 48.4% | 24.4% | 30.7% | 70.4% | 95.8% | 99.6% | 99.9% | 97.9% | 51.8% |
| 2018 | 68.1% | 53.0% | 27.6% | 32.4% | 67.5% | 95.6% | 99.6% | 99.7% | 98.5% | 53.9% |
| 2019 | 88.7% | 61.7% | 32.8% | 32.4% | 63.2% | 94.2% | 99.4% | 99.8% | 99.2% | 56.1% |

**B**

| Year | Age group | | | | | | | | | Total |
|------|------|-------|-------|-------|-------|-------|-------|-------|------|-------|
|      | 10< | 10-19 | 20-29 | 30-39 | 40-49 | 50-59 | 60-69 | 70-79 | 80≤ |       |
| 2009 | 1.5% | 7.4% | 21.8% | 18.2% | 7.6% | 1.6% | 1.5% | 0.9% | 1.0% | 11.8% |
| 2010 | 1.0% | 3.4% | 7.2% | 6.1% | 3.5% | 0.7% | 0.8% | 0.6% | 0.0% | 4.3% |
| 2011 | 0.2% | 2.1% | 3.1% | 2.5% | 1.9% | 0.5% | 0.0% | 0.8% | 0.8% | 2.0% |
| 2012 | 0.1% | 1.0% | 0.8% | 0.4% | 0.5% | 0.2% | 0.2% | 0.5% | 0.8% | 0.5% |
| 2013 | 0.0% | 0.3% | 0.5% | 0.3% | 0.5% | 0.2% | 0.2% | 0.2% | 0.0% | 0.3% |
| 2014 | 0.0% | 0.6% | 0.4% | 0.5% | 0.7% | 0.2% | 0.3% | 0.2% | 0.8% | 0.5% |
| 2015 | 0.0% | 0.8% | 0.6% | 0.4% | 0.4% | 0.3% | 0.3% | 0.3% | 0.0% | 0.4% |
| 2016 | 0.0% | 0.9% | 1.1% | 1.2% | 1.3% | 0.3% | 0.1% | 0.4% | 0.7% | 1.0% |
| 2017 | 0.1% | 0.6% | 1.2% | 0.9% | 1.2% | 0.4% | 0.4% | 1.0% | 0.3% | 0.9% |
| 2018 | 0.5% | 0.4% | 0.6% | 0.6% | 1.3% | 0.8% | 0.3% | 0.8% | 0.9% | 0.7% |
| 2019 | 0.6% | 1.3% | 2.9% | 4.3% | 6.5% | 2.7% | 1.3% | 1.1% | 1.1% | 3.7% |

**Fig 1. 2009–2019 changes in the positive rate of hepatitis A antibody in South Korea, by age.** (A) Seroprevalence of anti-HAV IgG. (B) Seroprevalence of anti-HAV IgM. Over the 2009–2019 period, the positive rate of anti-HAV IgG antibodies in teenagers has been continuously increasing, whereas the positive rate of antibodies in individuals in their 40s has decreased. Antibody titers in their 20s and 30s gradually increased or decreased. However, there was no significant change compared with

other age groups, and the antibody price was still low. In 2009, the anti-HAV IgM antibody showed high antibody titer, but gradually decreased and then increased suddenly in 2019. In 2009, high antibody titers were shown at all ages from teens to 40s. In particular, the IgM antibody value was high in individuals in their 20s and 30s. However, in 2019, 10 years later, high IgM antibody titers were observed in individuals in their 30s and 40s.

IgM is 1.2 or higher, and 0.8–1.2 is set as the gray zone. However, the difference in the rate of positivity between the two criteria was not significant, and therefore, 0.8 was considered as positive. In 2009, high rates were found in individuals in their teens up to their 40s, especially in those in their 20s (21.8%) and 30s (18.2%). As with anti-HAV IgG, males were found to have significantly higher positive rates than females (4.6% vs. 3.7%), and the positive rates tended to be higher in general hospitals. Anti-HAV IgM rates were high in the Incheon, Daejeon, Gyeonggi, Chungnam, and Chungbuk regions in 2009. Over 2009–2019, the total positive rate has been low in Busan, Gyeongnam, Gyeongbuk, Ulsan, Daegu, and Daejeon, but has been consistently higher in Incheon (8.9%) and Chungnam (5.9%) (Fig 2).

Since 2009, the antibody-positive rate of IgM declines sharply, increasing again at the time of the outbreak in 2019. During the outbreak in 2009, the hepatitis A antibody-positive rate was high in various regions, particularly in Seoul, Incheon, Daejeon, Gyeonggi, Gangwon, and Chungnam. However, in 2019, the antibody-positive rate is highest in the Incheon, Daejeon, Gyeonggi, Chungnam, and Sejong regions. SE, Seoul; BS, Busan; DG, Daegu; IC, Incheon; GJ, Gwangju; DJ, Daejeon; US, Ulsan; GG, Gyunggi; GW, Gangwon; CB, Chungbuk; CN, Chungnam; GB, Gyungbuk; GN, Gyungnam; JB, Jeonguk; JN, Jeonnam; JJ, Jeju; SJ, Seojong.

**Table 2. Anti-HAV IgG-positive rate in children younger than 10 years.**

| Years | Samples | Age Group (Months) | | | | | | | | | | Total |
|---|---|---|---|---|---|---|---|---|---|---|---|---|
| | | <12 | 12–24 | 24–36 | 36–48 | 48–60 | 60–72 | 72–84 | 84–96 | 96–108 | 108–120 | |
| 2009 | No. | 0 | 10 | 4 | 1 | 3 | 1 | 2 | 7 | 3 | 5 | 36 |
| | (%) | (0.0) | (62.5) | (66.7) | (25.0) | (100.0) | (20.0) | (100.0) | (87.5) | (75.0) | (62.5) | (62.1) |
| 2010 | No. | 2 | 105 | 32 | 49 | 39 | 31 | 33 | 48 | 34 | 28 | 401 |
| | (%) | (0.1) | (46.7) | (43.8) | (80.3) | (81.3) | (83.8) | (76.7) | (75.0) | (61.8) | (52.8) | (59.9) |
| 2011 | No. | 7 | 82 | 41 | 87 | 58 | 77 | 69 | 61 | 65 | 64 | 611 |
| | (%) | (63.6) | (33.2) | (43.2) | (75.7) | (71.6) | (87.5) | (82.1) | (65.6) | (70.7) | (62.7) | (60.6) |
| 2012 | No. | 8 | 111 | 39 | 75 | 81 | 76 | 79 | 80 | 79 | 58 | 686 |
| | (%) | (40.0) | (40.2) | (45.9) | (80.6) | (81.8) | (81.7) | (87.8) | (74.1) | (83.2) | (67.4) | (65.6) |
| 2013 | No. | 22 | 157 | 17 | 82 | 64 | 57 | 72 | 45 | 36 | 68 | 620 |
| | (%) | (40.7) | (41.6) | (54.8) | (89.1) | (77.1) | (86.4) | (77.4) | (73.8) | (66.7) | (70.1) | (61.5) |
| 2014 | No. | 16 | 124 | 11 | 61 | 52 | 31 | 39 | 44 | 44 | 48 | 470 |
| | (%) | (33.3) | (45.6) | (84.6) | (87.1) | (89.7) | (83.8) | (79.6) | (81.5) | (78.6) | (81.4) | (65.6) |
| 2015* | No. | 25 | 71 | 38 | 50 | 40 | 33 | 36 | 45 | 39 | 50 | 427 |
| | (%) | (33.3) | (36.6) | (92.7) | (87.7) | (95.2) | (86.8) | (90.0) | (84.9) | (76.5) | (82.0) | (65.5) |
| 2016 | No. | 19 | 54 | 36 | 26 | 34 | 32 | 37 | 33 | 46 | 53 | 370 |
| | (%) | (22.9) | (61.4) | (94.7) | (100.0) | (97.1) | (91.4) | (84.1) | (78.6) | (75.4) | (75.7) | (70.9) |
| 2017 | No. | 19 | 55 | 27 | 24 | 34 | 25 | 28 | 37 | 27 | 41 | 317 |
| | (%) | (27.5) | (62.5) | (96.4) | (96.0) | (85.0) | (75.8) | (84.8) | (86.0) | (73.0) | (80.4) | (70.9) |
| 2018 | No. | 14 | 43 | 18 | 14 | 14 | 17 | 17 | 22 | 33 | 41 | 233 |
| | (%) | (25.5) | (55.1) | (90.0) | (87.5) | (87.5) | (94.4) | (94.4) | (95.7) | (76.7) | (75.9) | (68.3) |
| 2019 | No. | 0 | 254 | 241 | 220 | 224 | 254 | 279 | 354 | 377 | 431 | 2,634 |
| | (%) | (0.0) | (75.6) | (97.6) | (99.5) | (96.1) | (97.7) | (96.2) | (92.9) | (83.0) | (82.6) | (86.5) |

* The year in which hepatitis A vaccination was included in the National Immunization Program in South Korea

| Year | Regions | | | | | | | | | | | | | | | | | Total |
|------|------|------|------|------|------|------|------|------|------|------|------|------|------|------|------|------|------|------|
| | SE | BS | DG | IC | GJ | DJ | US | GG | GW | CB | CN | GB | GN | JB | JN | JJ | SJ | |
| 2005 | 12.5% | 4.2% | 3.8% | 14.1% | 4.7% | 4.2% | 2.9% | 13.0% | 6.0% | 10.1% | 7.1% | 2.6% | 2.8% | 2.1% | 1.8% | 3.2% | | 8.7% |
| 2006 | 18.6% | 4.0% | 4.3% | 13.2% | 9.4% | 18.1% | 6.9% | 18.4% | 8.2% | 11.1% | 8.0% | 3.7% | 2.8% | 6.1% | 7.1% | 7.6% | | 12.0% |
| 2007 | 11.9% | 5.9% | 4.2% | 15.1% | 12.2% | 13.0% | 7.5% | 13.7% | 27.7% | 15.5% | 8.3% | 2.5% | 4.0% | 4.1% | 10.0% | 7.2% | | 10.0% |
| 2008 | 21.2% | 3.7% | 5.0% | 37.3% | 26.1% | 14.9% | 0.0% | 22.0% | 12.0% | 15.3% | 13.1% | 4.4% | 3.8% | 7.2% | 7.9% | 8.8% | | 15.9% |
| 2009 | 12.0% | 2.5% | 5.8% | 33.6% | 6.3% | 16.9% | 4.0% | 19.7% | 9.6% | 11.8% | 16.9% | 5.8% | 3.6% | 7.4% | 6.4% | 10.3% | | 11.5% |
| 2010 | 5.5% | 1.1% | 2.9% | 8.1% | 0.9% | 13.8% | 2.1% | 6.7% | 7.7% | 9.7% | 10.1% | 2.2% | 2.0% | 3.2% | 3.5% | 6.1% | | 4.7% |
| 2011 | 2.6% | 0.5% | 1.2% | 8.1% | 1.0% | 2.4% | 8.1% | 4.1% | 5.5% | 3.2% | 4.6% | 0.7% | 1.8% | 2.4% | 4.4% | 4.0% | | 2.4% |
| 2012 | 0.8% | 0.1% | 0.6% | 1.0% | 0.3% | 0.0% | 0.0% | 0.9% | 1.4% | 0.8% | 1.6% | 0.3% | 0.2% | 1.3% | 0.0% | 1.2% | | 0.5% |
| 2013 | 0.6% | 0.1% | 0.5% | 0.8% | 1.3% | 0.4% | 0.1% | 0.2% | 1.6% | 0.7% | 1.2% | 0.1% | 0.5% | 0.4% | 0.0% | 0.3% | | 0.3% |
| 2014 | 0.6% | 0.1% | 0.6% | 3.0% | 0.3% | 0.0% | 0.1% | 0.4% | 2.6% | 0.9% | 2.0% | 0.1% | 0.7% | 1.0% | 2.3% | 2.3% | | 0.5% |
| 2015 | 0.3% | 0.1% | 0.7% | 7.8% | 1.2% | 0.9% | 0.3% | 0.4% | 0.5% | 3.2% | 2.7% | 0.2% | 0.2% | 0.7% | 0.6% | 1.9% | 0.0% | 0.4% |
| 2016 | 1.0% | 0.3% | 0.7% | 6.6% | 2.0% | 2.3% | 1.3% | 0.9% | 2.1% | 1.6% | 3.7% | 0.3% | 1.7% | 1.5% | 4.0% | 4.3% | 9.7% | 1.0% |
| 2017 | 1.4% | 0.2% | 0.6% | 7.0% | 1.0% | 0.2% | 0.5% | 0.8% | 0.8% | 1.2% | 3.6% | 0.2% | 0.8% | 1.1% | 1.1% | 3.6% | 0.0% | 0.8% |
| 2018 | 1.1% | 0.3% | 0.6% | 1.8% | 0.6% | 1.5% | 0.9% | 0.9% | 1.1% | 1.3% | 1.9% | 0.4% | 0.5% | 0.4% | 0.0% | 1.5% | 5.7% | 0.7% |
| 2019 | 4.5% | 1.2% | 1.5% | 6.3% | 2.2% | 11.6% | 0.9% | 5.3% | 4.0% | 3.7% | 8.0% | 1.4% | 1.1% | 3.5% | 1.5% | 3.6% | 9.3% | 3.7% |

**Fig 2. Regional anti-HAV IgM antibody-positive rate changes over 2009–2019.**

During the second analysis period (2019), 211,629 cases were analyzed. The total positive rate was 3.69%, which was higher than the average positive rate in the previous 10 years. The highest positive rate was in individuals in their 40s (6.5%), and the lowest rate of 0.6% was observed in those younger than 10 years. The positive antibody rate tended to decrease in individuals in their 50s and older; however, the positive rate increased in those older than 80 years (Table 1, Fig 1B). The positive rate of anti-HAV IgM was higher in males (4.9%) than in females (3.2%), and general hospitals had the highest rate among the medical institutions (6.3%). Daejeon (11.6%) had the highest rate among the regions, and the neighboring regions Chungnam (8.0%) and Sejong City (9.3%) also had high rates (Fig 2).

## The relationship between antibody changes and patient reporting

We investigated the relationship between changes in the number of patient reports and anti-HAV IgM rates identified via laboratory tests since 2011, when the "all-patient report" was initiated. It was observed that the changes tended to be similar between the number of new patient outbreak reports and anti-HAV IgM-positive rates. When the complete patient report was started, the changes were found to be different between the IgM-positive rate and the tendency to report new patients, but they gradually became consistent. Therefore, it appears that the patient outbreak reporting system for hepatitis A in South Korea has been established appropriately (Fig 3A). During the second survey period, there was an outbreak of hepatitis A. The epidemic of 2019 began in January and declined sharply around September after measures were taken to prevent the population from eating food presumed to be responsible for the spread of infectious diseases. The positive rates of anti-HAV IgM tended to be similar in number to the number of patient reports, but they tended to be higher than the number of patient reports at the beginning (January) and end (December) of the epidemic (Fig 3B).

## The seasonality of anti-HAV antibodies

There were no significant differences in the positive monthly rates of anti-HAV IgG during the first analysis period of 10 years. During the 2019 epidemic, the total rate was higher than the average for the 2009–2018 period, but the monthly difference was not significant. However, during the first analysis period, the positive rate of anti-HAV IgM gradually increased

**A**

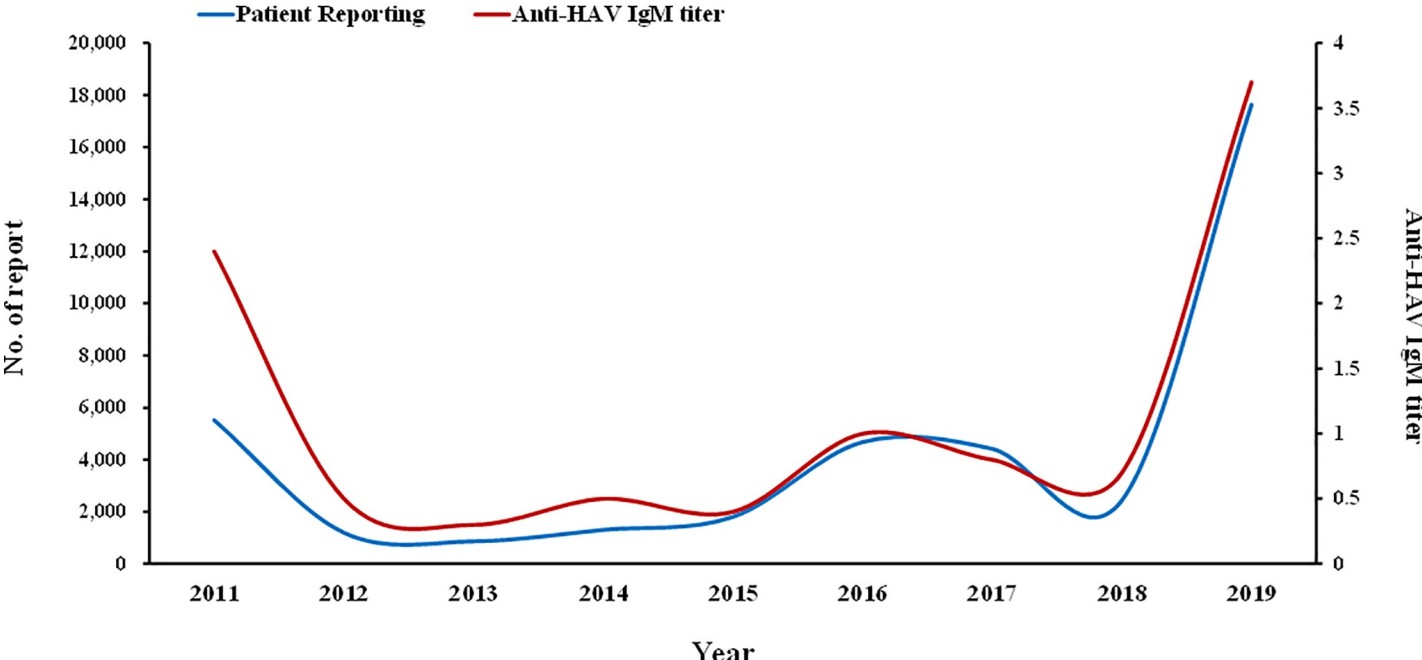

**B**

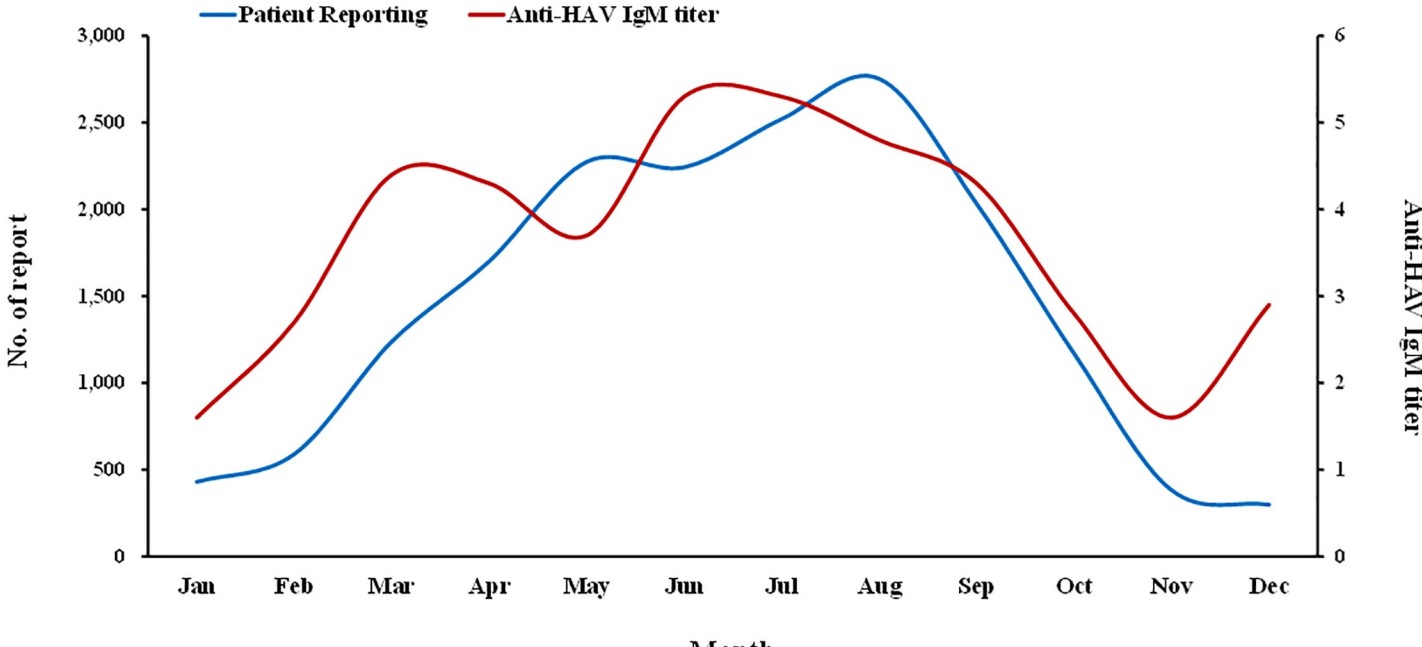

**Fig 3. Relationship between patient outbreak reports and anti-HAV IgM.** (A) Patient outbreak reports and anti-HAV IgM antibody-positive rates by year. (B) Monthly patient outbreak reports and anti-HAV IgM antibody-positive rates for 2019. The changes in the anti-HAV IgM antibody-positive rates and patient outbreak reports are similar since 2011 when the complete patient investigation began. From 2015, the trends in the two sets of data are consistent. However, in 2019, when there was an outbreak, the monthly patient report and the results of antibody levels from the five laboratories did not exactly match.

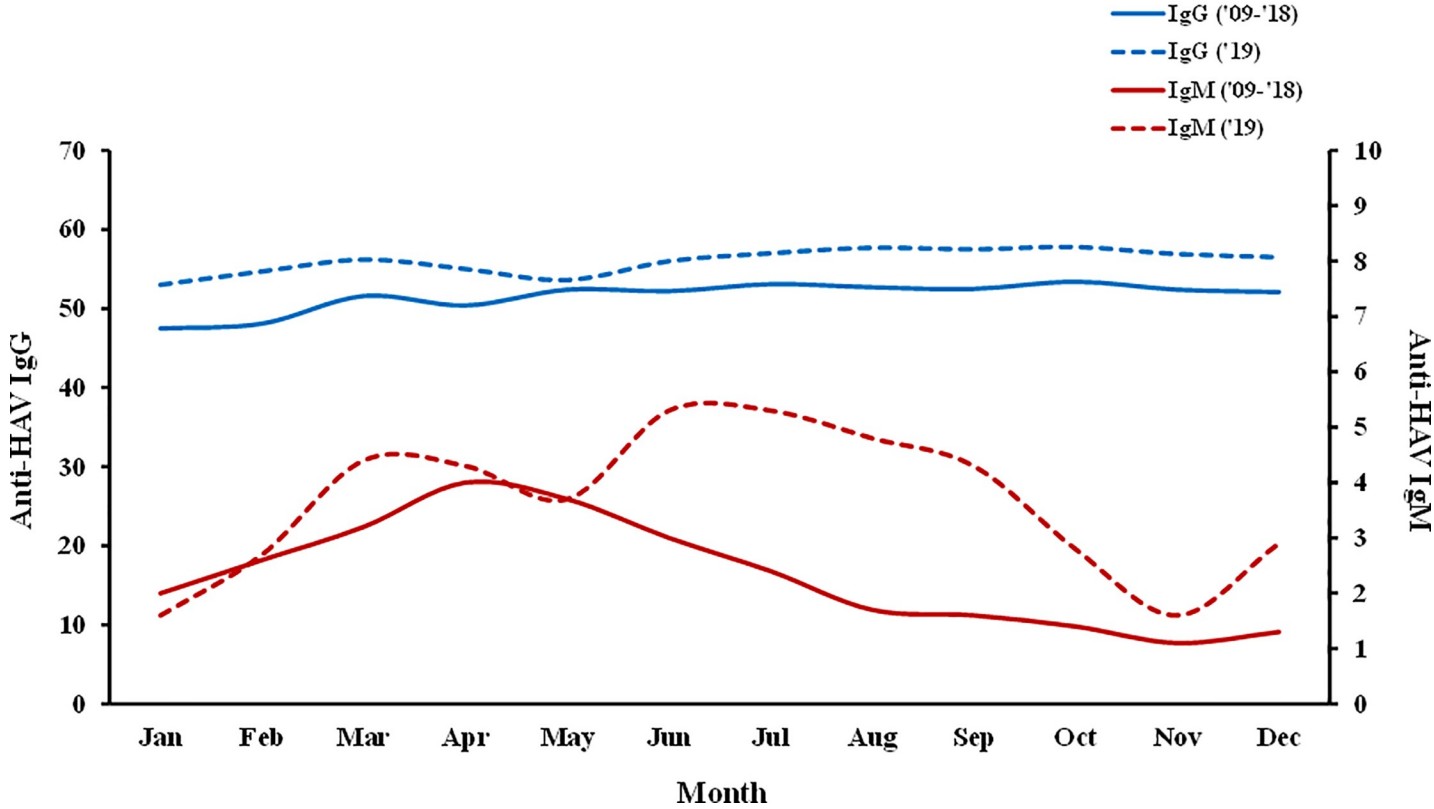

**Fig 4. Seasonality of the anti-HAV antibody-positive rate.**

from January to March, resulting in a high positive rate from March to June and a low positive rate after July. In 2019, unlike in the previous years, the positive rate increased rapidly until March, decreased briefly in May, increased again, and then decreased after September (Fig 4).

The anti-HAV IgG rate does not change significantly by month or year, but the anti-HAV IgM rate increases gradually from January to the highest antibody-positive rate in April. In 2019, there was no significant change in the antibody-positive rate of IgG. However, IgM exhibits a high positive rate in September, which appears to be a change brought about by the epidemic that was caused by new infectious agents.

## Discussion

During the first study period, the positive rate of the IgG antibody was 51.8% on average, but there were some differences by year. However, based on other data, the antibody-positive rates were in the range of 61.0% to 68.2% from 2005 to 2014. Additionally, from 2009 to 2014, the antibody-positive rates ranged from 61.0% to 63.8% and decreased in 2014. The antibody-positive rate investigated in other studies was the total anti-HAV rate that did not distinguish between the IgG and IgM rates. Anti-HAV IgM antibody titer results that were received from the same data source revealed that the rates were maintained at high levels ranging from 8.7% to 15.7% in the period from 2005 to 2009 and that they decreased to less than 1% after 2011 [7, 9]. We believe that this decrease in the antibody-positive rate of IgM resulted in the decrease in the total antibody-positive rate observed in other studies. In other words, it appears that the

difference in the antibody-positive rate depends on whether the method of measuring the total immunoglobulin concentration or that of measuring IgG and IgM rates separately is used. Therefore, we consider that a correction would be required when interpreting the results of tests performed when the antibody type to be investigated, while examining the antibody titer, is different. In addition, according to another report, the average antibody prevalence of domestic hepatitis A was approximately 69.06% from 2010 to 2014. This study surveyed health checkup data of individuals in their 20s and older and found that 76.4% of the screened subjects were 35–54 years old, indicating a higher proportion of individuals with higher antibody titers in these age groups compared with other age groups [8].

To minimize the scope for errors in our analysis, the total number of tests, including those of suspected patients and those conducted during health checkups, was included in the analysis. Although the data were not corrected based on the age, region, or population, the errors were minimized by increasing the number of analytical targets. The data analyzed in 2019 were from 596,245 patients, accounting for approximately 1% of the South Korean population. In 2019, the antibody-positive rate of children younger than 10 years increased rapidly.

In 2009, teenagers and individuals in their 20s were classified as being hepatitis A risk groups with a low IgG antibody-positive rate, and in fact, many patients who were reported were in their 20s [7, 9]. It was considered that the antibody was retained through natural immunity in the case of people in their 40s or older, and immunity was ensured in teenagers and younger children by vaccination. However, teenagers and individuals in their 20s had fewer opportunities to attain natural immunity due to improvements in personal hygiene and because of limited opportunity to receive a vaccination. Analysis of the change in the antibody-positive rate by year showed that the rates in younger children tended to increase gradually but that the rates in teenagers increased rapidly from 22.3% to 61.7%. Individuals in their 20s exhibited a slight increase, those in their 30s maintained an antibody titer of 30%, and the rate of those in their 40s tended to decrease. Based on the data from 10 years later, individuals with the lowest antibody titer rates for hepatitis A were in their 20s and 30s, indicating that although the risk groups did not change, the age of the risk groups simply increased.

The anti-HAV IgM rate increased from January and reached the highest positive rate in April, but the anti-HAV IgG rate did not change significantly by year or month. Considering the incubation period of hepatitis A, there may have been several infections around February and March. According to the results of a survey conducted in the United States, the person-to-person transmission rate of hepatitis A is 25%, the rate of infection resulting from unexplained causes is approximately 40% to 50%, the rate of cases infected by contaminated food is approximately 2%, and infection by water is extremely rare [10, 11]. This is because the source of infection disappears at the time of symptom onset, which makes it difficult to trace infection sources, such as food or water. In South Korea, however, February to March is the dry season, and groundwater sometimes becomes contaminated and concentrated; therefore, water-borne/foodborne diseases, such as norovirus infection, frequently occur [12]. In addition, there have been many cases involving hepatitis A outbreaks in South Korea that occurred in group residential facilities that use groundwater in the winter or spring. In the neighboring countries of Japan and China, cases involving hepatitis A outbreaks have mainly been reported in spring [13]. In 2019, there was an unusual case of a hepatitis A epidemic that persisted until fall because the population was continuously exposed to the HAV via imported salted shellfish (Fig 4). In other words, if there is no influence of contaminated food or other external sources of infection, it can be deemed that the main propagation route of domestic hepatitis A is likely to be contaminated water.

The first policy put forward by the government to reduce the incidence of hepatitis A is the new patient reporting system. This was introduced in order to be quickly and accurately

informed about the occurrence of infection in new patients and thus prevent further transmission of the infectious agents. Since 2011, there has been consistency between the anti-HAV IgM changes and number of patient reports; therefore, it appears that the patient reporting system has been properly established.

Another national policy is the introduction of national immunization programs for children. It is recommended that individuals be inoculated with the vaccination for the first time at 12 months of age and for the second time 6 months later, following which the antibodies would be produced after the individual reaches 24 months of age. According to the results of children younger than 10 years, more than 90% of the antibody-positive rates were observed in individuals who were 24 months and older since 2015, when the vaccination program was introduced (Table 2). In the meantime, it appears that the management of hepatitis A has been successful to some extent. However, a risk group with a low antibody-positive rate still exists, and hepatitis A occurs periodically in the spring. Thus, additional policy considerations are still required to further lower the risk of hepatitis A.

## Supporting information

**S1 Data.**
(XLSX)

## Acknowledgments

We thank the Seoul Clinical Laboratory, Samkwang, Green Cross, Seegene and Eawon medical foundation for providing the motivation and supporting data in this study.

## Author Contributions

**Conceptualization:** Deog-Yong Lee, Myung-Guk Han.

**Methodology:** Su-Jin Chae, Seung-Rye Cho, Chang-Ki Kim.

**Supervision:** Myung-Guk Han.

**Writing – original draft:** Deog-Yong Lee.

**Writing – review & editing:** Wooyoung Choi, Chang-Ki Kim.

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
