## [Decision Letter · Decision Letter 0]

12 Oct 2020

PONE-D-20-19554

Nationwide seroprevalence of hepatitis A in South Korea from 2009 to 2019

PLOS ONE

Dear Dr. Han,

I apologize for the delay in giving you a decision. One reviewer who agreed to review never responded to reminders and my personal message. To prevent further delay, I read your paper carefully and make a decision based on my own assessment and that of one reviewer. I agree with the reviewer and I like your work which will prove useful. I feel that more detailed description of the methods and better discussion of the results should suffice to satisfy us in the next round of review. Some discussion on herd immunity and the need and strategy to vaccinate specific groups (e.g. school children) might also be useful.   

After careful consideration, we feel that it has merit but does not fully meet PLOS ONE’s publication criteria as it currently stands. Therefore, we invite you to submit a revised version of the manuscript that addresses the points raised during the review process.

We look forward to receiving your revised manuscript.

Kind regards,

Dong-Yan Jin

Academic Editor

PLOS ONE

Journal Requirements:

2. In your ethics statement in the manuscript and in the online submission form, please provide additional information about the patient records used in your retrospective study. Specifically, please ensure that you have discussed whether all data were fully anonymized before you accessed them and/or whether the IRB or ethics committee waived the requirement for informed consent. If patients provided informed written consent to have data from their medical records used in research, please include this information.

3. Please discuss the possible limitations of this study.

4. Thank you for stating the following in the Competing Interests/Financial Disclosure* (delete as necessary) section:

'The authors have declared that no competing interests exist.'

We note that one or more of the authors are employed by a commercial company: 'Seoul Clinical Laboratories'.

'This study was supported by The Operation of Infectious Diseases Standard Laboratory

374 (4800-4837-301) and The Prevention of Infectious Diseases (4800-4838-303) in Korea CDC.'.

i) We note that you have provided funding information that is not currently declared in your Funding Statement. However, funding information should not appear in the Acknowledgments section or other areas of your manuscript. We will only publish funding information present in the Funding Statement section of the online submission form.

ii) Please remove any funding-related text from the manuscript and let us know how you would like to update your Funding Statement. Currently, your Funding Statement reads as follows:

'The authors received no specific funding for this work.'.

Additional Editor Comments: See above.

Reviewers' comments:

Reviewer's Responses to Questions

**Comments to the Author**

1. Is the manuscript technically sound, and do the data support the conclusions?

Reviewer #1: No

2. Has the statistical analysis been performed appropriately and rigorously? 

Reviewer #1: No

3. Have the authors made all data underlying the findings in their manuscript fully available?

Reviewer #1: Yes

4. Is the manuscript presented in an intelligible fashion and written in standard English?

Reviewer #1: Yes

5. Review Comments to the Author

Reviewer #1: The authors analyzed the seroprevalence of HAV in South Korea from 2009 to 2019. A huge amount of data was analyzed for this study, which should bring relevant information. There are some concerns however on the validity of the data used.

Major comments:

1. A concern of this study is the eventual variability in sensitivity and specificity among the different tests to detect anti-HAV antibodies in the different centers and period of times. The authors should discuss this issue.

2. The other concern is the validity of the data to represent the real HAV seroprevalence in each age group.

3. The discussion in line 296: based on other data: What does it mean? All the discussion should be revsed to provide a clear message from this data.

Minor comments

4. Abstract, line 31: remove at and use parenthesis for the percent.

5. Introduction: other liver comorbidities/co-infections (like hepatitis C) are also important risk factors for severity of this disease and should be mentioned in the Introduction.

6. Table 1: There is no need to include the percent negative in the table

7. Page 9, line 181: substitute dramatic by significant.

8. Fig. 1, 2 and 3: the legends cannot be a presentation of results, and the real legend is lacking. What is the meaning of cases? The absence of real legends in the figures make them difficult to understand.

6. PLOS authors have the option to publish the peer review history of their article (what does this mean?). If published, this will include your full peer review and any attached files.

Reviewer #1: No

---

## [Author Response · Author response to Decision Letter 0]

14 Dec 2020

November 13, 2020

Dong-Yan Jin

Academic Editor

PLOS ONE

Dear Prof. Jin,

We wish to resubmit the manuscript titled “Nationwide seroprevalence of hepatitis A in South Korea from 2009 to 2019.” The manuscript ID is PONE-D-20-19554.

We appreciate your consideration and the expedited review of the manuscript. We thank you and the reviewer for your thoughtful suggestions and insights. The manuscript has benefited from these insightful suggestions.

We have revised the manuscript according to the requirements of PLOS ONE. Furthermore, the manuscript has been rechecked and the necessary changes have been made in accordance with the reviewer’s suggestions. We also reviewed the manuscript with the help of experts. Nevertheless, if there is any part of the manuscript that does not meet the requirements of PLOS ONE policy, please let me know and I will revise the manuscript as soon as possible. The responses to all comments have been prepared and given below.

This study was supported by the Operation of Infectious Diseases Standard Laboratory (4800-4837-301) and the Prevention of Infectious Diseases (4800-4838-303) in Korea CDC. There are no conflicts of interest to declare. The funder provided support in the form of salaries for one of the authors (C.-G. Kim) but did not have any additional role in the study design, data collection and analysis, decision to publish, or preparation of the manuscript. The specific roles of these authors are articulated in the “Author Contributions” section of the submission system.

Thank you for your consideration. I look forward to working with you and the reviewer to move this manuscript closer to publication in the PLOS ONE.

Sincerely,

Myung-Guk Han, DVM, PhD

Division of Viral Diseases

Bureau of Infectious Diseases Diagnosis Control

Korea Disease Control and Prevention Agency

Osong-eup, Heungdeok-gu, Cheongju-si

Chungcheongbuk-do 28159, Republic of Korea

Tel.: +82-43-719-8190

Fax: +82-43-719-8219

E-mail: mghan@korea.kr

A. Journal Requirements:

1. PLOS ONE's style requirements

Response: As per recommendations, we have modified the manuscript according to the PLOS ONE style requirement. The title page has been modified according to the guidelines. In the acknowledgment section, the part that could be misunderstood as referring to a budget has been revised to the project name. The picture file was converted into individual tiff files, according to guidelines.

2. Ethic Statement

Response: The recommended statement as well as additional information was added in the online submission form.

3. The possible limitation of this study

Response: The limitations of this paper were mentioned in the Discussion (Line 315-320). The data used in this paper were from a group that was tested for hepatitis A and were not corrected for age, region, or population ratio. Therefore, we tried to minimize errors that may occur by analyzing as many samples as possible, and the percentage of samples represented up to 1% of the total population of South Korea.

4. The Competing Interests/Financial Disclosure

Response: The content has been revised, as recommended, and the statement was added in the letter above.

5. Acknowledgments Section

Response: This study was conducted as a result of a surveillance project conducted by the Centers for Disease Control and Prevention. We have corrected what might be misunderstood as a reference to the budget.

6. Ethics

Response: The statement on ethics has already been mentioned in the Materials and Methods section. The ethical approval statement given before the references has been deleted.

B. Reviewer’s comments

Reviewer #1: The authors analyzed the seroprevalence of HAV in South Korea from 2009 to 2019. A huge amount of data was analyzed for this study, which should bring relevant information. There are some concerns however on the validity of the data used.

Major comments:

1. A concern of this study is the eventual variability in sensitivity and specificity among the different tests to detect anti-HAV antibodies in the different centers and period of times. The authors should discuss this issue.

Response: The reviewer raised important points. The data we analyzed are the results of tests performed by five major Korean clinical testing institutions. As mentioned in lines 108-115 of the text, automated equipment was used for all evaluations. These devices have been registered as in vitro diagnostic medical devices and are used worldwide. In addition, these institutions have undergone quality evaluation and received quality certification from a well-known institution. We believe that the results used in this paper have sufficient sensitivity and specificity, as verified by the quality certification. Hence, these data were used for research.

2. The other concern is the validity of the data to represent the real HAV seroprevalence in each age group.

Response: We also pondered on the same concern, and we agree with the reviewer’s opinion. When selecting a group for actual research, the representativeness of the group will always be a problem. Hence, we discussed the problem in the first discussion (lines 264-282). Depending on sample group selection and test method, studies in the same group may have different results. Hence, to minimize these errors and derive representative results, we analyzed close to 1% of the Korean population (lines 271-274). Therefore, we believe that we have produced data that are close to the representative values of antibody titers by age.

3. The discussion in line 296: based on other data: What does it mean? All the discussion should be revised to provide a clear message from this data.

Response: “Other data” refers to the data used when we were setting up the sample group. In particular, these are based on the papers cited in line 272. These data also addressed the previous comment on sample selection. Please let me know if the explanation is unclear, and we will further modify it for clarity.

Minor comments

4. Abstract, line 31: remove at and use parenthesis for the percent.

Response: This part has been modified according to the reviewer’s suggestion.

5. Introduction: other liver comorbidities/co-infections (like hepatitis C) are also important risk factors for severity of this disease and should be mentioned in the Introduction.

Response: We discussed this in the Introduction according to the reviewer’s suggestion.

6. Table 1: There is no need to include the percent negative in the table

Response: We have deleted the percentage of negative from the table according to the reviewer’s suggestion. 

7. Page 9, line 181: substitute dramatic by significant.

Response: We have revised the word dramatic to significant according to the reviewer’s suggestion.

8. Fig. 1, 2 and 3: the legends cannot be a presentation of results, and the real legend is lacking. What is the meaning of cases? The absence of real legends in the figures make them difficult to understand. 

Response: The results were supplemented in the legends of Figures 1, 2, and 3 according to the reviewer’s suggestion

---

## [Decision Letter · Decision Letter 1]

23 Dec 2020

Nationwide seroprevalence of hepatitis A in South Korea from 2009 to 2019

PONE-D-20-19554R1

Dear Dr. Han,

We’re pleased to inform you that your manuscript has been judged scientifically suitable for publication and will be formally accepted for publication once it meets all outstanding technical requirements.

Kind regards,

Dong-Yan Jin

Academic Editor

PLOS ONE

Reviewers' comments:

Reviewer's Responses to Questions

**Comments to the Author**

1. If the authors have adequately addressed your comments raised in a previous round of review and you feel that this manuscript is now acceptable for publication, you may indicate that here to bypass the “Comments to the Author” section, enter your conflict of interest statement in the “Confidential to Editor” section, and submit your "Accept" recommendation.

Reviewer #1: All comments have been addressed

2. Is the manuscript technically sound, and do the data support the conclusions?

Reviewer #1: Yes

3. Has the statistical analysis been performed appropriately and rigorously? 

Reviewer #1: Yes

4. Have the authors made all data underlying the findings in their manuscript fully available?

Reviewer #1: Yes

5. Is the manuscript presented in an intelligible fashion and written in standard English?

Reviewer #1: Yes

6. Review Comments to the Author

Reviewer #1: The authors addressed the comments, one by one in the letter of response, and edited accordingly the manuscript.

7. PLOS authors have the option to publish the peer review history of their article (what does this mean?). If published, this will include your full peer review and any attached files.

Reviewer #1: No

---

## [Editor Report · Acceptance letter]

27 Jan 2021

PONE-D-20-19554R1 

Nationwide seroprevalence of hepatitis A in South Korea from 2009 to 2019 

Dear Dr. Han:

I'm pleased to inform you that your manuscript has been deemed suitable for publication in PLOS ONE. Congratulations! Your manuscript is now with our production department. 

Kind regards, 

on behalf of

Professor Dong-Yan Jin 

Academic Editor

PLOS ONE